# Mechanisms of PIEZO Channel Inactivation

**DOI:** 10.3390/ijms241814113

**Published:** 2023-09-14

**Authors:** Zijing Zhou, Boris Martinac

**Affiliations:** 1Victor Chang Cardiac Research Institute, Lowy Packer Building, Darlinghurst, NSW 2010, Australia; z.zhou@victorchang.edu.au; 2St Vincent’s Clinical School, University of New South Wales, Darlinghurst, NSW 2010, Australia

**Keywords:** mechanosensitive channels, mechanopathologies, force-from-lipids, TMEM150C, STOML3, MDFIC, MDFI

## Abstract

PIEZO channels PIEZO1 and PIEZO2 are the newly identified mechanosensitive, non-selective cation channels permeable to Ca^2+^. In higher vertebrates, PIEZO1 is expressed ubiquitously in most tissues and cells while PIEZO2 is expressed more specifically in the peripheral sensory neurons. PIEZO channels contribute to a wide range of biological behaviors and developmental processes, therefore driving significant attention in the effort to understand their molecular properties. One prominent property of PIEZO channels is their rapid inactivation, which manifests itself as a decrease in channel open probability in the presence of a sustained mechanical stimulus. The lack of the PIEZO channel inactivation is linked to various mechanopathologies emphasizing the significance of studying this PIEZO channel property and the factors affecting it. In the present review, we discuss the mechanisms underlying the PIEZO channel inactivation, its modulation by the interaction of the channels with lipids and/or proteins, and how the changes in PIEZO inactivation by the channel mutations can cause a variety of diseases in animals and humans.

## 1. Introduction

Mechanosensitive PIEZO ion channels are evolutionarily conserved membrane proteins whose function is critical for normal physiology in living cells and organisms [1,2,3,4] ranging from single-celled ciliated protozoans to multicellular organisms, including plants, insects, worms, and humans [1,2,3,4,5]. In humans, they play a key role in sensing touch, tactile pain, breathing, and blood pressure. However, they differ in their expression patterns and functions. The PIEZO1 channel is present in non-sensory tissues, with particularly high expression in the lung, bladder, and skin. In contrast, the PIEZO2 channel is predominantly present in sensory tissues, such as dorsal root ganglia (DRG) sensory neurons and Merkel cells [5,6]. While PIEZO1 channels are activated by the force-from-lipids indicating that they are inherently mechanosensitive [7,8], the inherent mechanosensitivity has not yet been demonstrated for PIEZO2 [9]. In addition to mechanical stimuli, PIEZO channels are also powerfully modulated by voltage [10,11]. Voltage modulation may be explained by the presence of an inactivation gate in the pore. Mutations that cause human diseases, such as Xerocytosis [12,13], affect the channel inactivation and profoundly shift voltage sensitivity of the PIEZO1 channels towards the resting membrane potential and strongly promote voltage gating [10].

Inactivation is a general property of most types of ion channels enabling filtering out repetitive or prolonged stimuli by blocking the flow of ions via a mechanism other than the closing of the channel [13,14,15,16,17]. Structural studies reveal diverse mechanisms for inactivation, while most of the inactivation processes involve conformational changes in one or multiple inactivation gates located within the pore region. This normally happens upon opening of the channels and generally limits the size of the permeation path for permeating ions. The inactivation state usually slowly recovers by transitioning to a closed channel state, which allows the channels to go through the open-inactive-closed cycle again before the next stimulus. Importantly, inactivation can be modified by intrinsic or extrinsic factors, with the latter including changes in pH, voltage, or temperature, surrounding membrane lipids, and cellular components binding to the channels [5,18,19].

The focus of this article is on PIEZO1 and PIEZO2 mechanosensitive channels, which play a key role in sensing touch, tactile pain, stretch induced by lung inflation, or blood pressure [5]. However, the two channels differ in their expression patterns and functions. In addition to their inherent mechanosensitivity [7,8], they can be modulated by the force-from-filament transmitted to the channels by cytoskeletal or extracellular matrix molecules [18,20,21,22]. Cryo-EM characterization of both channels reveals their very similar triskelion-like structure [23,24,25,26]. A functional PIEZO channel is composed of three homomeric subunits with 38 transmembrane domains each, acting as an extensive blade-like element to transduce force to the pore. The C-terminus of the protein is composed of an extracellular cap domain, the pore-forming inner transmembrane helix (IH) and outer transmembrane helix (OH), and an intracellular C-terminal domain. Both PIEZO1 and PIEZO2 possess voltage-dependent inactivation [27,28,29,30,31,32,33,34]. The whole-cell patch clamp with indentation shows a more rapid inactivation for PIEZO2, which has an inactivation time constant of around 5 ms compared to PIEZO1 of around 15 ms at a holding potential of −80 mV [29,35]. Interestingly, only Piezo1 but not Piezo2 shows a fast inactivation inward current in the cell-attached mode [36]. Previous studies have shown that the C terminus of the PIEZO channels bears critical amino acids that are essential for their inactivation [28,30,37]. However, the precise mechanism underlying the inactivation of the channels remains elusive. In the present article, we summarize the current knowledge about the PIEZO channel inactivation with a structural insight. We also explain how the interacting partners, including lipids and proteins, exert their effects on this PIEZO channel gating property. Furthermore, we highlight our recent studies on a novel family of cellular proteins that tightly interact and regulate the inactivation of these channels (Table 1). Finally, we discuss how the inactivation of both ion channels is linked to human diseases and give some thought to what future studies on PIEZO channel inactivation may unveil.

## 2. Intrinsic Mechanisms Underlying Inactivation of the PIEZO Channels

PIEZO1 and PIEZO2 exhibited fast inactivation in HEK293T or N2A cells during hyperpolarizing membrane potentials in outside-out and whole-cell patch clamping modes, while both channels were inactivated slower at depolarizing membrane potentials [6,35,47]. PIEZO2 was further found to contribute to the rapid-inactivating, endogenous MS current in mouse DRG neurons [48]. Since then, several disease-causing, gain-of-function (GOF) mutants have provided critical insights into the potential mechanisms of PIEZO1 and PIEZO2 inactivation. For PIEZO1, mutations that lead to a single amino acid substitution at human PIEZO1 M2225 (M2225R) or R2456 (R2456H), or mouse PIEZO1 M2241 or R2482, slow down PIEZO1 inactivation and are linked to xerocytosis (Figure 1A) [12,13]. A more comprehensive genetic screening identified other mutants, including R1358P, A2020T, T2127M, and E2496ELE, with all of them reducing PIEZO1 inactivation [49].

Further studies on how the two amino acids M2225 and R2456 affect PIEZO1 inactivation reveal that these two are working independently but also cooperatively, while mutating both amino acids almost completely removed the inactivation of the channel (Figure 1A) [13]. For PIEZO2, mutations at human PIEZO2 E2727 (E2727del) or I802 (I802F) exert similar effects as they alter PIEZO2 inactivation kinetics and cause distal arthrogryposis in humans, which can be recapitulated in gene-modified mice [50]. This information provided critical insights into the potential mechanisms of the PIEZO channel inactivation while falling short of elucidating the mechanisms fully without having access to a 3D structure of the channels back then.

Thanks to the development of the high-resolution cryo-electron microscopy (cryo-EM) techniques within the last ten years, the structures of PIEZO1 and PIEZO2 have been solved at a resolution of ≥3.7 Å [24,25,26,51] (Figure 1B). Since then, inactivation-associated mutants were mapped back to the structures and the underlying mechanisms could be explained in more detail. For example, the M2225 and R2456 residues were found to belong to the C-terminal extracellular (cap) and IH regions, respectively [30]. It was also found that the cap and IH regions largely contribute to the inactivation of both PIEZO1 and PIEZO2 [28]. Furthermore, exchanging the cap regions of PIEZO1 and PIEZO2 revealed that the cap region accounts for the difference between the gating kinetics of the two channels, therefore the cap region must be important for inactivation [31]. In support of this idea, pulling amino acids within the cap region with magnetic nanoparticles largely reduced PIEZO1 inactivation [52]. A more precise dissection of the cap region confirmed that several subdomains within the cap of PIEZO2 were sufficient to confer the rapid inactivation of PIEZO2 to PIEZO1 [30]. Unlike the cap region, for which our understanding of how inactivation is affected remains unclear, our understanding of the inactivation mechanism within the inner helix has been clarified at a single amino acid level [43]. Two positively charged residues, which are K2453 and R2456 in human PIEZO1 or K2479 and R2482 in mouse PIEZO1, were found to be essential for PIEZO1 voltage-dependent inactivation. Mutating R2482 largely removes inactivation while the remaining inactivation still exhibits voltage dependence. The size rather than the charge of the residue seems to yield the effect, as R2482K or R2482Q both slow down inactivation significantly (Moroni et al., 2018) [11]. In addition, the effect of R2482 is conserved in PIEZO2 as mutating the homologous mouse PIEZO2 R2756 to histidine (R2756H) or lysine (R2756K) slows down the inactivation of a PIEZO1/PIEZO2 chimera [34]. On the other hand, neutralizing the positive charge on K2479 by mutating it to glutamine (K2479Q) or reverting the charge by mutating it to glutamic acid (K2479E) abolished the voltage dependence of PIEZO1 inactivation [11]. Importantly, they exerted opposite effects on inactivation, as K2479Q enhanced while K2479E removed inactivation, indicating the charge of this residue is essential for voltage dependence. Moreover, L2475 and V2476 residues residing close to K2479 and R2582 in mouse PIEZO1, or L2749 and V2750 in the mouse PIEZO2, regulate the inactivation of PIEZO1 and PIEZO2 in a conserved way (Figure 1C). Mutating both sites to serine (L2475S/V2476S or L2750S/V2751S) starkly removed inactivation in both channels. The structure of PIEZO1 indicates a 10 Å radius of pore size at L2475 and V2476. Thus, it is hypothesized that these two amino acids switch angles and face toward the pore, consequently narrowing down the pore size to <6 Å radius and forming a hydrophobic barrier that leads to inactivation [28]. Other structural motifs of PIEZO also contribute to inactivation. For example, the proximal intracellular C terminal domain (CTD) was shown to bear two very narrow constrictions at mouse PIEZO1 M2493/F2494 and E2537/P2536, with the MF constriction mildly contributing to PIEZO1 inactivation but not PIEZO2. Our recent data also indicate the involvement of the anchor domain in PIEZO1 inactivation, since the addition of two glycine residues to the G2163 residue in the anchor domain-outer helix of human PIEZO1 largely removed the inactivation of the channel [14]. It is worth mentioning that motifs outside the C terminus can influence channel inactivation as well [32]. First, a number of the gain-of-function mutants that reduce PIEZO channel inactivation, are in the peripheral transmembrane domains. However, it remains unclear what is the underlying mechanism. Second, tissue-specific alternative splicing, which removes the PIEZO2 peripheral intercellular region, alters PIEZO2 inactivation kinetics [53]. We reason that the inactivation of PIEZO channels may be affected by peripheral regions due to the force sensing or transmission, while the main inactivation gate is most likely located within the ECD cap domain and the inner helix pore-forming transmembrane domain.

## 3. Extrinsic Factors Modifying PIEZO Inactivation

Most of our understanding of the intrinsic mechanisms of the PIEZO channel inactivation, as described above, is based on overexpressing PIEZO channels in a heterologous cell system. However, given that the activity of the channels can be modulated by their surrounding microenvironment, various extrinsic factors can affect the channel kinetics, which is consistent with the fact that in some native cell lines, PIEZO1 does not inactivate rapidly [31,54] (Figure 2A). In the following section, we focus on what is currently known about the modulation of the PIEZO channel inactivation by extrinsic factors, including voltage, temperature, membrane lipids, or intracellular proteins, which is summarized in Table 1.

The PIEZO channel inactivation exhibits strong voltage dependency, which seems to be associated with the charged residues aligned within the pore-forming inner helix [10,11]. As we have addressed in the previous section, although extrinsic, membrane potential is closely linked to the intrinsic channel structure by affecting the movement of the electrical charges associated with the channel conformational changes. Another extrinsic factor is pH, for which it has been found that protonation stabilizes the inactivation of PIEZO1 [19]. This finding is based on the observation that PIEZO1 currents upon multiple pressure pulses decrease with decreasing pH. Changes in pH do not influence the gain-of-function mutant R2456H or double-mutant M2225R/R2456K, which lack inactivation already, further supporting the idea that low pH promotes desensitization by altering the inactivation of the channel. In addition, temperature regulates the inactivation of both channels across different species, as lowering the temperature slows down inactivation for PIEZO1 and PIEZO2 overexpressed in a heterologous system or endogenous mechanosensitive currents in mouse DRG or duck TG cells. Mechanisms for temperature affecting inactivation are not yet clear; however, the stiffness of the lipid membrane, which is generally affected by temperature, seems to be not involved [38].

Lipids as integral components of the plasma membrane or signaling molecules are also actively regulating the PIEZO channel inactivation [18]. Margaric acid inhibits both PIEZO1 and PIEZO2 channel activity but not the inactivation kinetics. On the other hand, linoleic acid (LA) 18:2 slows down PIEZO1 and PIEZO2 inactivation while potentiating both channels [38,39]. The effect has been established by incubating cells that endogenously express PIEZO1 or PIEZO2 such as HMVEC or MCC13 cells, or N2A cells overexpressing PIEZO channels with LA followed by whole-cell patch clamping. This can be partially explained by LA increasing membrane disorder and, therefore, altering membrane physical properties, as LA decreases the lipid-melting temperature, which also sensitizes MscL overexpressed in N2A cells. Alterations in membrane properties by polyunsaturated fatty acids (PUFAs) on PIEZO1 and PIEZO2 channels determine the time course of the channel inactivation, as arachidonic acid (AA) 20:4 and eicosapentaenoic acid (EPA) 20:5 enhances while docosahexaenoic acid (DHA) 22:6 reduces the inactivation of PIEZO1 [39]. In contrast, EA or DHA do not influence PIEZO2 channel inactivation. This suggests a specific role of PUFAs in regulating PIEZO1 activity by possibly occupying lipid-binding pockets within the channel. Nevertheless, LA and EPA have been shown to have the potential for treating PIEZO-related LOF or GOF diseases in mouse models [38,40]. Ceramide is also implied to regulate PIEZO1 inactivation in freshly isolated second-order mesenteric artery endothelial cells (MAECs) [42]. Endogenous PIEZO1 current does not inactivate in the MAECs. By inhibiting SMPD3, which is a neutral sphingomyelinase that catalyzes the transition of sphingomyelin into ceramide, the native PIEZO1 current gains rapid inactivation. This can be rescued by incubating the cells with ceramide, which restores the non-inactivating character of the PIEZO1 currents. Ceramide was also shown to be essential for MAECs’ response to flow, as the channels stay open for longer periods under continuous flow stimulation. While it is essential for the biophysical properties of PIEZO1 in the endothelium, ceramide fails to regulate PIEZO1 overexpressed in a heterologous system. This suggests that ceramide may be required, but it is not sufficient to expand the inactivation gate for which other components in the native endothelial cell are cooperatively functioning to regulate inactivation. In addition, our previous studies revealed a role of cholesterol and PIP_2_ in regulating PIEZO1 inactivation [45]. Cholesterol fostered PIEZO1 clustering when it is overexpressed in the HEK cells [41]. Removing cholesterol with methyl-β-cyclodextrin (MBCD) largely removed PIEZO1′s inactivation and lowered the PIEZO1 gating threshold to pressure. MD simulations identified multiple possible binding regions to cholesterol in PIEZO1 and provided insight into a possible mechanism for cholesterol-regulating PIEZO1 through specific binding to the channel. Interestingly, PIP_2_ is also suggested to bind to PIEZO1. A conserved motif in PIEZO1 K2166-K2169 is a potential binding site for PIP_2_ based on our simulations while deleting these four lysine residues removed PIEZO1 inactivation [45].

Importantly, PIEZO1 and PIEZO2 have been reported to interact physically and functionally with other cellular proteins. Consequently, these interacting partners potentiate or inhibit PIEZO channel activity through different mechanisms. Examples include SERCA2 binding to and reducing PIEZO1 and PIEZO2 peak current; E-cadherin binding to and potentiating PIEZO1 and PIEZO2 channel activity; STOML3 sensitizing PIEZO1 and PIEZO2 to mechanical stimuli; MTMR2 inhibiting PIEZO2 activity through PIP2; Polycystin-2 (PKD2) interacting with PIEZO1 and reducing its function; PECAM-1 interacting with PIEZO1 at cell junctions and suppressing PIEZO1′s activity; TRPM4 interacting with PIEZO1 and amplifying PIEZO1 dependent calcium signals in cardiomyocytes; and TMEM150C interacting with PIEZO2 and positively regulating PIEZO1, PIEZO2, and TREK-1 channel activity [18,56,57,58,59,60]. Those proteins that exert influence on PIEZO channels but not through direct interaction with the channel are not discussed here. It is worth noting that, along with potentiating or reducing the mechano-activated (MA) currents of PIEZO1 and PIEZO2, some of the binding partners, such as E-cadherin or PKD2, also regulate inactivation kinetics of the PIEZO channels; however, compared to the influence on the peak current, their effect on inactivation seems not to be the primary cause but rather a consequence of altered channel activity. One binding partner that more specifically regulates PIEZO inactivation is TMEM150C, which was first thought to be an independent mechanosensitive channel that confers a relatively slow inactivating MA current [61]. It was later shown that the slow inactivated MA current was the PIEZO1 current, as TEME150C overexpression did not produce MA currents in PIEZO1 knockout cells [43]. The stark change in PIEZO1 inactivation kinetics with co-expression of TMEM150C led to further studies, which revealed an inactivation-removing effect of TMEM150C on PIEZO1, PIEZO2, and TREK-1 across multiple species, indicating that TMEM150 is a general modulator of the mechanosensitive channels [43]. It is hypothesized that TMEM150C produces such an influence on MS channels by regulating the composition and physical properties of the membrane.

We recently identified MDFI and MDFIC, which belong to the MyoD family of inhibitors as a novel family of proteins that physically interact with PIEZO1 and PIEZO2 by using affinity purification of endogenously expressed PIEZO1 from fibroblasts [55]. We were able to resolve a high-resolution structure of the PIEZO1-MDFIC complex by cryo-EM that clearly showed how a cysteine-rich, palmitoylated C terminus of MDFIC is inserted into the pore region of PIEZO1 (Figure 2B). MDFIC and MDFI seem to strongly regulate PIEZO1 and PIEZO2 inactivation kinetics since the co-expression of MDFIC or MDFI completely removed the inactivation of PIEZO channels in a heterologous system (Figure 2C) while knocking down or mutating MDFIC restored a fast-inactivating endogenous PIEZO1 current in fibroblasts. We further found that the C-terminal palmitoylation is essential for the function of MDFIC interaction with both PIEZO channels. Furthermore, molecular dynamics simulations indicated that the palmitate chains may physically interfere with the amino acids located at the putative inactivation gate within the pore-forming helix of PIEZO1. To our knowledge, this is the first structure showing the PIEZO channel complex with a binding partner. At the same time, our finding provides important information for a better understanding of the PIEZO channel inactivation process.

## 4. Altered Inactivation Kinetics of PIEZO Channels Is Related to Human Diseases

Given the ubiquitous expression and unique function of PIEZO channels, their mutations can be expected to cause severe consequences for mechanosensory transduction in a living organism. A large number of molecular mechanisms could explain disease-causing PIEZO channel mutations with abnormal inactivation kinetics being a common observation [62,63,64]. For example, malfunction in PIEZO1 inactivation leads to hereditary Xerocytosis (or dehydrated hereditary stomatocytosis, DHS) [12,13]. DHS is characterized by fragile, dehydrated red blood cells, which results in hemolysis and severe anemia. A possible mechanism for PIEZO1 involvement in DHS is the lack of inactivation in the gain-of-function PIEZO1 mutants causing calcium overload in red blood cells (RBCs). The abundantly expressed Ca^2+^-dependent KCa_3.1_ channels are subsequently activated leading to excessive outflow of potassium, which further leads to dysregulation of ion concentration and osmolarity in the RBCs [65]. On the other hand, genetic screening has located two mutations in PIEZO2 that cause Distal Arthrogryposis Type 5 (DA5) in humans [66]. Typical clinical symptoms of DA5 can be described as generalized autosomal dominant contractures. PIEZO2 mutants associated with DA5 have slower inactivation kinetics or faster recovery from inactivation. Deleting the mouse PIEZO2 E2799 residue, which mimics the human PIEZO2 mutant E2727del, is sufficient to cause DA5-like syndromes in mice. The mechanism for PIEZO2 E2727del is thought to be the hyperactivity of PIEZO2 in the proprioceptive sensory neurons, which can be partially rescued by a neural transmission blocker botulinum toxin or EPA that restores PIEZO2 inactivation [40].

In addition, the proper function of PIEZO1 was shown to be essential for lymphatic valve formation and development of the lymphatic system, given that lymphatic endothelial cell (LEC)-specific PIEZO1 knockout mice exhibit postnatal lethality due to abnormal development of the lymphatic valve [67]. In humans, loss-of-function mutants of PIEZO1 cause general lymphatic dysplasia (GLD), which is characterized by severe lymphoedema affecting the whole body [63,68]. The loss-of-function in these PIEZO1 mutants causing the disease has been attributed to aberrant protein trafficking and stability, or the lack of channel activity. However, most of the PIEZO LOF mutants have not been studied in detail by the patch clamp. Unlike the removal of inactivation in the GOF mutants, possibilities of enhanced inactivation in PIEZO channels and their disease-causing roles are less concerning partially because of the already rapid inactivation of the channels when they are overexpressed. In contrast, many native cells including the endothelia cells exhibit a non-inactivating PIEZO1 current. Our findings have demonstrated that MDFIC and MDFI contribute to a slowly inactivating PIEZO1 in the native cells. Strikingly, knocking out MDFIC causes early postnatal death for the mice resulting from the abnormal development of lymphatic valves. The coincidence of almost identical phenotypes in MDFIC and LEC-specific PIEZO1 knockout reminds us of a plausible mechanism for GLD, which is the insufficient removal of inactivation of PIEZO1 due to the lack of MDFIC/MDFI expression or binding [55,69]. Those LOF mutants within the interacting interface of PIEZO1 and MDFIC, such as PIEZO1 V2171F, are therefore of high interest to be investigated in the future.

## 5. Conclusions and Expectations

The inactivation of PIEZO1 and PIEZO2 ion channels has emerged as a pivotal area of research with far-reaching implications for our understanding of mechanotransduction in various physiological and pathological processes. Both ion channels play essential roles in converting mechanical forces into electrical and biochemical signals, thereby influencing numerous cellular and tissue functions.

As crucial players in various sensory systems, PIEZO1 and PIEZO2 channels have provided valuable insights into the mechanisms of touch and mechanical sensation, highlighting the importance of these channels in our ability to perceive and respond to the external environment. Dysfunctional PIEZO channels have been implicated in several pathological conditions, such as familial dehydrated stomatocytosis, congenital joint contractures, chronic pain syndromes, and cardiovascular disorders. Understanding their inactivation mechanisms could pave the way for targeted therapeutic interventions to alleviate these conditions by modulating mechanosensory responses and may also shed light on tissue morphogenesis and regeneration processes. Consequently, a detailed understanding of PIEZO channel inactivation mechanisms may inspire the design of novel drugs that can mimic or interfere with these processes.

In conclusion, the study of PIEZO1 and PIEZO2 channel inactivation and associated discoveries holds great promise for advancing our understanding of mechanosensation and its impact on various (patho)physiological processes. The implications of these discoveries extend to therapeutic interventions for diverse mechanopathologies. By delving deeper into the mechanisms of inactivation, researchers will be able to unlock new avenues for drug development and gene editing technologies, and thus potentially revolutionize the field of mechanobiology.

## Figures and Tables

**Figure 1 ijms-24-14113-f001:**
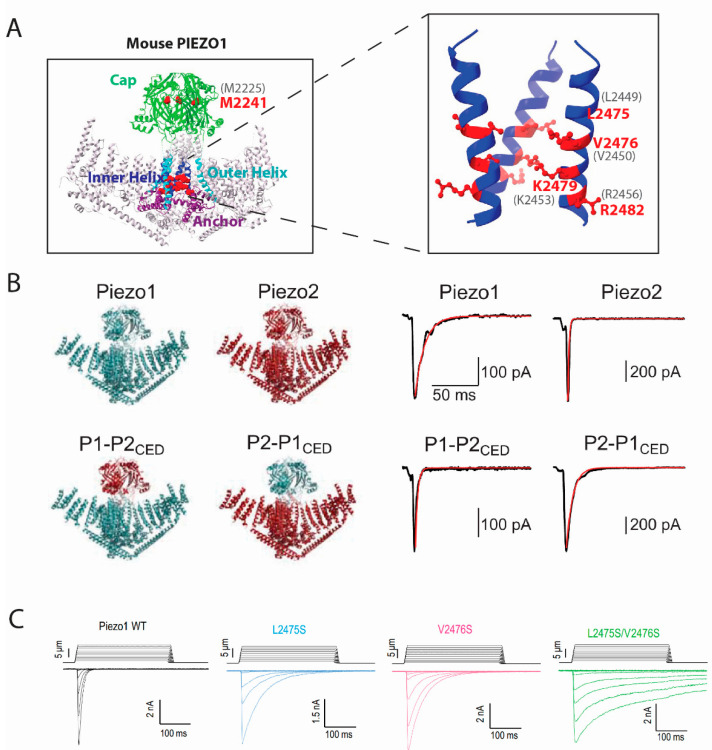
**Essential domains and residues for PIEZO inactivation.** (**A**) Cap domain and inner helix are highlighted in the 3D structure of mouse PIEZO1 (PDB: 6BPZ). Critical amino acids in mouse PIEZO1, together with their positions, are highlighted in red; positions of amino acids in human PIEZO1 are in grey. The right panel presents a zoom-in diagram of the inner helix with the critical amino acids. (**B**) Structure overview of wild-type PIEZO1 and PIEZO2, or chimeric PIEZO1 or PIEZO2 fused with the ‘Cap’ domain (C terminal extracellular domain, CED) from each other, is shown in the left panel. Inactivation of PIEZO1 is faster than PIEZO2 (right panel, up). Swapping the CED exchanges the inactivation constant of the channels, as P1-P2_CED_ inactivates faster than the P2-P1_CED_ (right panel, down). The figure is adapted from (Wu et al., 2017) under CC BY-NC-ND 4.0. (**C**) Manipulating the hydrophobic gate by mutating L2475/V2476 of mouse PIEZO1 significantly reduces inactivation compared to the wild-type. Figure is adopted from [28] under CC BY-NC-ND 4.0.

**Figure 2 ijms-24-14113-f002:**
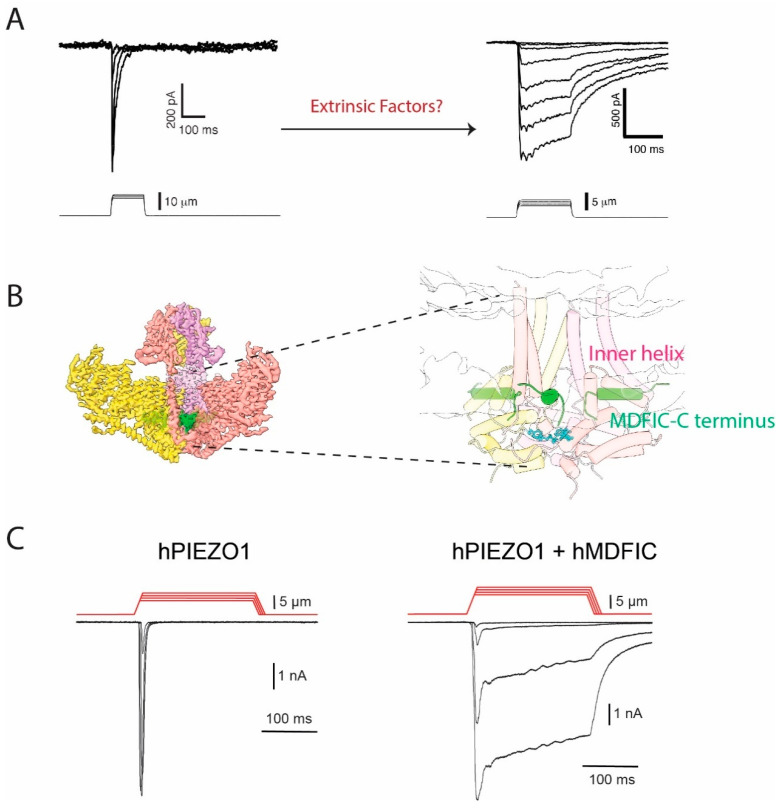
**Interacting proteins may explain the slow-inactivating PIEZO1 currents in native cells.** (**A**) Indentation-induced whole-cell currents in N2A cells overexpressing mouse PIEZO1 (Left) or native PIEZO1 current in the mouse embryonic stem cells (Right). Native PIEZO1 shows a slow inactivating kinetics. (**B**) 3D structure of PIEZO1-MDFIC Complex. The C terminus of MDFIC is shown in green. MDFIC inserts into PIEZO1′s pore module and stays near the inner helix, as detailed in the right panel. (**C**) HEK cells overexpressing PIEZO1 show a rapid inactivating current (Left). Upon co-expression of MDFIC, the inactivation is largely removed (Right). Figures are adopted from [54] and [55] under CC BY-NC-ND 4.0.

**Table 1 ijms-24-14113-t001:** Extrinsic factors affecting PIEZO1 and PIEZO2 inactivation.

Classification	Channel Type	Effect	Potential Mechanisms	References
**Environmental**				
Voltage	PIEZO1, PIEZO2	Slows down inactivation at depolarizing potential; enhances inactivation at hyperpolarizing potential	Possibly affects the charged amino acids at the inner helix of PIEZO1 and PIEZO2	[10,31,35]
Temperature	PIEZO1, PIEZO2	Colder temperature enhances inactivation of PIEZO channels	Changes membrane stiffness and modulates inactivation of PIEZO2; mechanisms on PIEZO1 is unknown.	[28]
pH	PIEZO1	Protonation enhances inactivation in PIEZO1	Unknown	[19]
**Lipids**				
linoleic acid (LA) 18:2	PIEZO1, PIEZO2	Slows down channels’ inactivation	Increases lipid membrane instability	[38,39]
arachidonic acid (AA) 20:4	PIEZO1, PIEZO2	Enhances channels’ inactivation	Exerts alterations of membrane properties combined with unknown direct protein interacting mechanisms	[38,39]
eicosapentaenoic acid (EPA) 20:5	PIEZO1 PIEZO2	Enhances channels’ inactivation	as above	[39,40]
docosahexaenoic acid (DHA) 22:6	PIEZO1	Reduces PIEZO1’s inactivation	as above	[39,41]
ceramide	PIEZO1	Important for maintaining the native slow inactivating PIEZO1 currents in ECs.	Possibly reduces the membrane curvature suggested by MD simulation	[42]
cholesterol	PIEZO1	Necessary for PIEZO1’s fast inactivation in the HEK cells	Possibly stiffens the membrane	[41,43,44]
PIP2	PIEZO1	Necessary for PIEZO1’s fast inactivation in the HEK cells	Binds to human PIEZO1 K2166-K2169 suggested by MD simulations. These four lysine residues are important for PIEZO1’s inactivation.	[45,46]
**Interacting Proteins**				
TMEM150C	PIEZO1, PIEZO2	Reduces PIEZOs’ inactivation	Unknown	[43]
MDFIC/MDFI	PIEZO1, PIEZO2	Removes PIEZOs’ inactivation	Inserts into the pore module of PIEZO1 and PIEZO2; palmitoylation on the C terminal cysteins interacts with essential residues in PIEZOs’ inner helix.	[37]

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
