# Peer review of "Mechanisms of PIEZO Channel Inactivation"

_ijms, 2023, doi:10.3390/ijms241814113_

Round 1

Reviewer 1 Report

The mini-review is valuable in its focus on one aspect of the function of extremely large and complicated PIEZO channels involved in human pathology, insightful and highlights recent developments in this complex area. However, before proceeding with publication, I believe a few revisions are necessary to enhance the clarity and accuracy of the content.

1.     Chapter 2: Amino Acid Mutations and Figure Correspondence: In this section, I noticed a discrepancy between the numbering of the amino acids mentioned in the text (2225, 2456, 2241, 2482) and those in Figure 1A (2241, 2475, 2476, 2479, 2582). This inconsistency can lead to confusion for readers trying to correlate information between the text and the figure. I recommend revising the text to ensure consistent numbering and referencing throughout this section.

2.     Figure 1A: Domain Structure Clarification: It has come to my attention that the domain structure of PIEZO channels, as depicted in Figure 1A, lacks clarity and fails to adequately represent important elements, such as the anchor domain-outer helix, which you discussed on page 4. To address this issue, consider revising the figure to accurately depict the various domains, helices, and clusters you mention in the text. This alignment will provide readers with a visual aid that corresponds seamlessly to your narrative.

3.     Additional Residue Positions in Figure 1A: Your manuscript highlights the significance of residues like those of the PIP2-binding cluster (2166-2169) on page 7. Integrating these important residue positions into Figure 1A will bolster the understanding of readers, allowing them to connect the written content with the visual representation more effectively.

4.     Inclusion of Domain Swap Experiment Description: I believe that  description and reference to the domain swap experiment depicted in Figure 1B are currently missing from your manuscript.

By addressing these issues, your manuscript will become more reader-friendly and better aligned with the valuable insights you have shared.

Minor issues:

page3 : " (Voltage gating mechanism of PIEZO channels, 2018)" - the title of the reference, instead of author name(s) is provided- please, correct this;

page 7: "This suggests a specific role of PUFAs on regulating PIEZO1 activity by possibly forming a lipid binding pocket within the channel" - This sentence should be changed -  lipids do not form pockets within channels but rather occupy them.

page 8: "palmitoylation chains" should be "palmitate chains"

I came across sentences that appear convoluted and are proving challenging to comprehend. To improve clarity, you might consider simplifying these sentences. At times, breaking a complex sentence into two can enhance understanding.

Specific issues:

page 2: "whole cell patch clamp"  change to "whole-cell patch clamp";

page 3: "which L2475 and V2476 residues residing....." "which" should be omitted;

page 4: "Our recent data also indicate the involvement of the anchor domain, for which adding two glycine residues at anchor domain-outer helix linking human PIEZO1G2163 residue largely removed inactivation of PIEZO1 (...)." - this sentence is unclear and confusing;

page 6: "pore forming inner helix" should be "pore-forming inner helix";

page 6:  "exhibits voltage-dependency" should be "exhibits voltage dependence";

page 8: "polycyctin-2" should be "polycystin-2"

Author Response

The mini-review is valuable in its focus on one aspect of the function of extremely large and complicated PIEZO channels involved in human pathology, insightful and highlights recent developments in this complex area. However, before proceeding with publication, I believe a few revisions are necessary to enhance the clarity and accuracy of the content.

  1. Chapter 2: Amino Acid Mutations and Figure Correspondence: In this section, I noticed a discrepancy between the numbering of the amino acids mentioned in the text (2225, 2456, 2241, 2482) and those in Figure 1A (2241, 2475, 2476, 2479, 2582). This inconsistency can lead to confusion for readers trying to correlate information between the text and the figure. I recommend revising the text to ensure consistent numbering and referencing throughout this section.

We thank the Reviewer for pointing out the mistake. One wrongly labelled amino acid, R2582, in the left panel of Figure 1.A has been corrected to R2482.

The differences in numbering of the amino acids in the text or figure are because of different versions (human or mouse) of PIEZO1 they are in. We detailed the positions of amino acids for both human and mouse PIEZO1 in the text, while in figure A, the numbering was based on a mouse version given the structure is for the mouse PIEZO1. To avoid the confusion, we have added the positions of amino acids for human PIEZO1 in grey to the figure.

  1. Figure 1A: Domain Structure Clarification: It has come to my attention that the domain structure of PIEZO channels, as depicted in Figure 1A, lacks clarity and fails to adequately represent important elements, such as the anchor domain-outer helix, which you discussed on page 4. To address this issue, consider revising the figure to accurately depict the various domains, helices, and clusters you mention in the text. This alignment will provide readers with a visual aid that corresponds seamlessly to your narrative.

Corrected accordingly.

  1. Additional Residue Positions in Figure 1A: Your manuscript highlights the significance of residues like those of the PIP2-binding cluster (2166-2169) on page 7. Integrating these important residue positions into Figure 1A will bolster the understanding of readers, allowing them to connect the written content with the visual representation more effectively.

Unlike the essential amino acids or defined domains of PIEZO1, which have been stringently validated by various groups, the PIP2 binding cluster has been indicated by computational simulation only. Therefore, in order not to overstate their significance, we decided not to highlight the cluster in our annotated structure but just discuss it in the text.

  1. Inclusion of Domain Swap Experiment Description: I believe that  description and reference to the domain swap experiment depicted in Figure 1B are currently missing from your manuscript.

We thank the reviewers for their comment. We have described cap domain swapping experiment in the text: ‘Furthermore, exchanging the cap regions of PIEZO1 and PIEZO2 revealed that the cap region accounts for the difference between the gating kinetics of the two channels therefore the cap region must be important for inactivation (Wu, Young et al. 2017)’.

In addition, citation of the original article has been added to the figure legend.

By addressing these issues, your manuscript will become more reader-friendly and better aligned with the valuable insights you have shared.

Minor issues:

page3 : " (Voltage gating mechanism of PIEZO channels, 2018)" - the title of the reference, instead of author name(s) is provided- please, correct this;

Corrected.

page 7: "This suggests a specific role of PUFAs on regulating PIEZO1 activity by possibly forming a lipid binding pocket within the channel" - This sentence should be changed -  lipids do not form pockets within channels but rather occupy them.

The sentence has been corrected and now reads: “This suggests a specific role of PUFAs on regulating PIEZO1 activity by possibly occupying lipid binding pockets within the channel.”

page 8: "palmitoylation chains" should be "palmitate chains"

Corrected.

Comments on the Quality of English Language

I came across sentences that appear convoluted and are proving challenging to comprehend. To improve clarity, you might consider simplifying these sentences. At times, breaking a complex sentence into two can enhance understanding.

Specific issues:

page 2: "whole cell patch clamp"  change to "whole-cell patch clamp";

Corrected.

page 3: "which L2475 and V2476 residues residing....." "which" should be omitted;

Corrected.

page 4: "Our recent data also indicate the involvement of the anchor domain, for which adding two glycine residues at anchor domain-outer helix linking human PIEZO1G2163 residue largely removed inactivation of PIEZO1 (...)." - this sentence is unclear and confusing;

The sentence has been rewritten to: “Our recent data also indicate the involvement of the anchor domain in the PIEZO1 inactivation, since addition of two glycine residues to the G2163 residue in the anchor domain-outer helix of the human PIEZO1 largely removed inactivation of the channel (Vero Li, C et al. 2021).”

page 6: "pore forming inner helix" should be "pore-forming inner helix";

Corrected.

page 6:  "exhibits voltage-dependency" should be "exhibits voltage dependence";

Corrected.

page 8: "polycyctin-2" should be "polycystin-2"

Corrected.

Reviewer 2 Report

In the manuscript “Mechanisms of PIEZO channel inactivation” Zhou and Martinac review the modulation of the mechanosensitive cation channels PIEZO1 and PIEZO2 by physical changes (voltage, temperature), pH, lipids, and proteins. These extrinsic factors, summarized in Table 1, either maintain or impair the fast inactivation of PIEZO channels. In addition, the authors give a nice overview on the intrinsic mechanisms of PIEZO channel gating and discuss altered PIEZO channel inactivation with respect to hereditary diseases. This review is highly topical and well written. I would only advise the following editorial changes:

(1) Some unnecessary spaces were found between words or words and punctuation marks.

(2) Table 1, first row: Line break within “Channels” should be avoided.

(3) Table 1, row “PIP2”: The reference “Activation of TRPV1 channels inhibits mechanosensitive Piezo channel activity by depleting membrane phosphoinositides, 2015” should be given as “Borbiro et al., 2015”.

(4) Reference list: The following reference is missing: Istvan Borbiro, Doreen Badheka, Tibor Rohacs. (2015) Activation of TRPV1 channels inhibits mechanosensitive Piezo channel activity by depleting membrane phosphoinositides. Sci Signal Feb 10;8(363):ra15. doi: 10.1126/scisignal.2005667.   

Author Response

In the manuscript “Mechanisms of PIEZO channel inactivation” Zhou and Martinac review the modulation of the mechanosensitive cation channels PIEZO1 and PIEZO2 by physical changes (voltage, temperature), pH, lipids, and proteins. These extrinsic factors, summarized in Table 1, either maintain or impair the fast inactivation of PIEZO channels. In addition, the authors give a nice overview on the intrinsic mechanisms of PIEZO channel gating and discuss altered PIEZO channel inactivation with respect to hereditary diseases. This review is highly topical and well written. I would only advise the following editorial changes:

(1) Some unnecessary spaces were found between words or words and punctuation marks.

Unnecessary spaces have been removed in the revised manuscript.

(2) Table 1, first row: Line break within “Channels” should be avoided.

“Channels” has been replaced with “Channel type”.

(3) Table 1, row “PIP2”: The reference “Activation of TRPV1 channels inhibits mechanosensitive Piezo channel activity by depleting membrane phosphoinositides, 2015” should be given as “Borbiro et al., 2015”.

The reference by Borbiro et al., 2015 has been included in the Table 1.

(4) Reference list: The following reference is missing: Istvan Borbiro, Doreen Badheka, Tibor Rohacs. (2015) Activation of TRPV1 channels inhibits mechanosensitive Piezo channel activity by depleting membrane phosphoinositides. Sci Signal Feb 10;8(363):ra15. doi: 10.1126/scisignal.2005667.

The reference by Borbiro et al., 2015 has been included in the Reference list.

Reviewer 3 Report

The reviw by Zhou, Z. and Martinac, B. about the inactivation mechanisms of Piezo channels appears well-organized and quite interesting. Authors performed a nice review, in my opinion. They could include more things, like the role of these channels in neoplasms or in fibroblastic response, but, honestly I am unable to find any significant fault, which is quite strange when reviewing any manuscript.

The text has a generally good grammar / style.

Author Response

The reviw by Zhou, Z. and Martinac, B. about the inactivation mechanisms of Piezo channels appears well-organized and quite interesting. Authors performed a nice review, in my opinion. They could include more things, like the role of these channels in neoplasms or in fibroblastic response, but, honestly I am unable to find any significant fault, which is quite strange when reviewing any manuscript.

The text has a generally good grammar / style.

We thank the reviewer for the suggestion to include more information on the role of the Piezo channels. However, this would be appropriate in a more general review discussing the role of the channels in cell pathophysiology, whereas this review is focusing exclusively on the inactivation mechanism in these ion channels.